# “Notame”: Workflow for Non-Targeted LC–MS Metabolic Profiling

**DOI:** 10.3390/metabo10040135

**Published:** 2020-03-31

**Authors:** Anton Klåvus, Marietta Kokla, Stefania Noerman, Ville M. Koistinen, Marjo Tuomainen, Iman Zarei, Topi Meuronen, Merja R. Häkkinen, Soile Rummukainen, Ambrin Farizah Babu, Taisa Sallinen, Olli Kärkkäinen, Jussi Paananen, David Broadhurst, Carl Brunius, Kati Hanhineva

**Affiliations:** 1Department of Clinical Nutrition and Public Health, University of Eastern Finland, 70210 Kuopio, Finland; stefania.noerman@uef.fi (S.N.); ville.m.koistinen@uef.fi (V.M.K.); marjo.tuomainen@uef.fi (M.T.); iman.zarei@uef.fi (I.Z.); topi.meuronen@uef.fi (T.M.); ambbab@student.uef.fi (A.F.B.); taisa.sallinen@uef.fi (T.S.); 2School of Pharmacy, University of Eastern Finland, 70210 Kuopio, Finland; merja.hakkinen@uef.fi (M.R.H.); soile.rummukainen@uef.fi (S.R.); olli.karkkainen@uef.fi (O.K.); 3Institute of Biomedicine, University of Eastern Finland, 70210 Kuopio, Finland; jussi.paananen@uef.fi; 4Centre for Integrative Metabolomics & Computational Biology, School of Science, Edith Cowan University, Joondalup, WA 6027, Australia; d.broadhurst@ecu.edu.au; 5Department of Biology and Biological Engineering, Chalmers University of Technology, 41296 Gothenburg, Sweden; carl.brunius@chalmers.se; 6Chalmers Mass Spectrometry Infrastructure, Chalmers University of Technology, 41296 Gothenburg, Sweden; 7Department of Biochemistry, Food Chemistry and Food Development unit, University of Turku, 20014 Turun yliopisto, Finland

**Keywords:** metabolomics, LC–MS, mass spectrometry, metabolic profiling, computational statistical, unsupervised learning, supervised learning, pathway analysis

## Abstract

Metabolomics analysis generates vast arrays of data, necessitating comprehensive workflows involving expertise in analytics, biochemistry and bioinformatics in order to provide coherent and high-quality data that enable discovery of robust and biologically significant metabolic findings. In this protocol article, we introduce notame, an analytical workflow for non-targeted metabolic profiling approaches, utilizing liquid chromatography–mass spectrometry analysis. We provide an overview of lab protocols and statistical methods that we commonly practice for the analysis of nutritional metabolomics data. The paper is divided into three main sections: the first and second sections introducing the background and the study designs available for metabolomics research and the third section describing in detail the steps of the main methods and protocols used to produce, preprocess and statistically analyze metabolomics data and, finally, to identify and interpret the compounds that have emerged as interesting.

## 1. Introduction

The rapid technical development of instrumentation for biomolecule analysis has led to a wide application of metabolomics in biological and biomedical research. Due to its very high sensitivity and the ability to concomitantly assess thousands of molecular features, liquid chromatography coupled with mass spectrometry (LC–MS) is making its way as the key analytical tool in the field of discovery-driven metabolic profiling [1,2,3]. The LC–MS platform generates large amounts of signals—biological signals from metabolites, their adducts, fragments, isotopes and instrument noise, thereby necessitating adequate computational tools to process, analyze and interpret the data [4,5]. Although the data processing solutions for complex metabolomics data are accumulating with increasing speed, they continue to be the bottleneck within the analysis, especially the metabolite identification process [6,7,8]. Starting from the acquisition of data to the identification of metabolites, the metabolic profiling workflow involves numerous steps that require expertise in analytical chemistry, biochemistry, bioinformatics and data analysis—click-and-go online tools may therefore not provide adequate reliability. To guarantee high quality output from metabolomics experiments, cooperation of scientists with various backgrounds and expertise is needed.

First, the production of high-quality metabolomics data requires high quality samples originating from studies with meaningful research questions, adequate sample preparation and know-how in operating MS instruments in order to get out the maximum performance of the sensitive measurements. The acquired data needs to undergo several preprocessing steps, starting from data collection (peak picking), where it is imperative to understand the detection threshold and signal-to-noise ratios of the measurement. This is then followed by a multi-step processing phase involving imputation, normalization, data reduction and clean-up, which determines the quality of the data that is used in downstream data-analysis, metabolite identification and biological interpretation of the results. All of these steps need to follow necessary quality assurance and quality control procedures for reliable outcome of the metabolomics analysis [9,10]. Finally, the compounds that have emerged as interesting in the given study setup need to be identified using a combination of automated metabolite identification algorithms and exploration of the raw LC–MS/MS spectral data.

Although the currently proposed non-targeted metabolic profiling workflow is applicable on basically any metabolomics study, it has been developed and utilized mainly on food and nutritional approaches. Therefore, examples provided here on the presentation of results are from studies within that field. In fact, food and nutrition sciences encompass a versatile array of research fields, which have adopted metabolomics as one of the most important analytical tools during the past decade [9]. For example, metabolic profiling allows a comprehensive analysis of the chemical composition of food and estimating the impact of industrial processing and modifications by gut microbiota [11,12]. Likewise, when assessing the actual health outcomes of certain diets or specific foods, metabolic profiling enables pointing out the areas of metabolism that are reflecting the dietary differences; especially when data are correlated with other, traditional clinical variables, they may raise novel hypotheses on the molecular-level linkage between diet and health [13,14,15].

Here, we present analytical workflows suitable for any non-targeted metabolic profiling study in a systematic manner (Figure 1), with a major focus on data-analysis challenges. We also present a new R package: notame (version 0.0.1, https://github.com/antonvsdata/notame), where we have bundled many of the data-analysis tools used in our lab so that they are easy to adopt for other scientists working in the field of metabolic profiling. This includes the pre-processing steps and visualizations in Section 3.2.2, Section 3.2.3, Section 3.2.4, Section 3.2.5, statistical tests and multivariate models in Section 3.3, as well as the visualizations in Section 3.4. The package documentation contains extensive instructions for using the package, along with a template script for preprocessing and analyzing data from a single-batch LC–MS experiment as well as a small example dataset.

## 2. Experimental Design

The non-targeted metabolic profiling analytical workflow presented here includes steps from sample preparation and LC–MS analysis all the way to metabolite identification (Figure 1). It is noteworthy to mention, however, that the study design and careful planning for the sampling are very important part of the study governing the quality of the results and therefore require special attention [9]. Herein, we focus on metabolomics analysis performed in one batch (where the number of samples typically reaches 200–300 samples). However, the procedures are in general applicable for larger, multi-batch experiments, although extra procedures for quality control are in order [10,16].

### 2.1. Materials

Sample preparation materials:a.96-well plate (Thermo Scientific, Rochester, NY, USA, Cat.No. 260252),b.Filter plate (Agilent, Santa Clara, CA, USA, Cat.No. A5969002)c.96-Well cap mats (Thermo Scientific, Roskilde, Denmark, Cat.No. 276002)d.Syringe filters (PALL Corporation, Ann Arbor, MI, USA, Cat.No. 4552T)e.Syringe Norm-Ject^®^ tuberculin 1 mL (Henke Sass Wolf, Tuttlingen, Germany, Cat.No 4010-200V0)f.Wide orifice pipette tips (Thermo Scientific, Vantaa, Finland, Cat.No. 9405050)g.Homogenizer microtubes (OMNI International, Kennesaw, GA, USA, Cat.No 19-620

LC–MS materials:h.Reversed-phase chromatography (RP) column: Zorbax Eclipse XDB-C18, particle size 1.8 µm, 2.1 × 100 mm (Agilent Technologies, Santa Clara, CA, USA, Cat.No. 981758-902).i.Hydrophilic interaction chromatography (HILIC) column: Acquity UPLC BEH Amide 1.7 µm, 2.1 × 100 mm (Waters Corporation, Milford, MA, USA, Cat.No. 186004801).

Reagents:a.Acetonitrile, ACN (HiPerSolv CHROMANORM, VWR Chemicals, Fontenay-sous-Bois, France, Cat.No. 83640.320)b.Methanol, MeOH (CHROMASOLV™ LC–MS Ultra, Riedel-de Haën™, Honeywell, Seelze, Germany, Cat.No. 14262-2L)c.Formic acid (Optima LC/MS, Fisher Chemical, Geel, Belgium, Cat.No. A117-50)d.Ammonium formate (CHROMASOLV™ LC–MS Ultra, Honeywell Fluka, Seelze, Germany, Cat.No. 14266-25G)e.Ultra-pure water (Class 1, ELGA PURELAB Ultra Analytical, Lane End, UK)

### 2.2. Equipment

The current workflow is demonstrated with one suitable LC–MS instrumentation and software combination but can likewise employ any other high-accuracy LC–MS setup.

Sample preparation and LC–MS instruments:a.Centrifuges: For 96-well plates: Heraus Megafuge 40R (ThermoFisher Scientific, Osterode, Germany), for microcentrifuge tubes: Centrifuge 5804R (Eppendorf, Hamburg, Germany)b.Vortex: Vortex Genie 2 (Scientific Industries, Bohemia, NY, USA)c.Homogenizer: Bead Ruptor 24 Elite with OMNI BR CRYO unit (OMNI International, Kennesaw, GA, USA)d.Shaker: Multi Reax (Heidolph, Schwabach, Germany)e.1290 Infinity Binary UPLC system (Agilent Technologies, Waldbronn, Karlsruhe, Germany)f.6540 UHD accurate-mass quadrupole-time-of-flight mass spectrometer (qTOF-MS) with Jetstream ESI source (Agilent Technologies, Santa Clara, CA, USA)

Software:g.Agilent MassHunter Acquisition B.07.00 (Agilent Technologies),h.MS-DIAL version 3.70 [17],i.MS-FINDER version 3.24 [18],j.R version 3.5.0 [19]k.Multiple Experiment Viewer (MeV) version 4.9.0 (http://mev.tm4.org/).

## 3. Analytical Procedure and Results

### 3.1. LC–MS Analysis

#### 3.1.1. Sample Preparation

Sample preparation for the non-targeted metabolite profiling work aimed to extract in a single attempt as wide range of metabolites as possible with, in general, minimal sample workup. Therefore, straightforward, simple extraction protocols were preferred. Protocol 1 was designed for extracting plasma/serum samples at a ratio of 1:5 with ACN and Protocol 2 for extracting homogenized tissue samples at a ratio of 1:6 with 80% methanol.

**Protocol** **1:**Plasma/Serum Samples

1.Thaw plasma/serum samples in ice water and keep them on wet ice during all the waiting periods.2.Place the 96-well plate on wet ice for sample preparation and set the filter plate on it.3.Add 400 μL of cold ACN to the filter plate well.4.Vortex a plasma/serum sample 10 s at the maximum speed.5.Add 100 μL of plasma/serum sample to the same well as ACN.6.To prepare the pooled quality control (QC) samples, collect 10 μL aliquots of each sample and add them to the same clean microcentrifuge tube and finally, mix properly.7.Mix ACN and sample by pipetting four times. Use wide orifice Finn Pipette tips to avoid tip clogging.8.Repeat steps 1–5 for all samples. Lastly, use the same procedure for the QC sample. For the extraction blank, perform step 3 (cold ACN without sample) and use the same procedure thereon.9.Filter the precipitated samples by centrifuging the plate for 5 min at 700× g at 4 °C.10.Remove the filter plate and seal the 96-well plate tightly with the 96-well cap mat to avoid sample evaporation.11.Analyze the samples immediately or store the plate at +4 °C for a maximum of 1 day or at −20 °C until analysis.

**Protocol** **2:**Tissue Samples

12.Weigh a maximum of 300 mg frozen tissue into 2-mL OMNI microtube with beads. Keep the samples on dry ice.13.Add ice cold 80% methanol in a ratio of 500 µL solvent per 100 mg tissue and keep the tubes on wet ice. Include an extraction blank with solvent only.14.**Optional step:** In the case of metabolite-dense sample material (e.g., plants), it might be necessary to use a more diluted solvent/sample ratio to avoid analytical problems, such as saturation of the detector or overloading of the column.15.Homogenize samples with a Bead Ruptor 24 Elite homogenizer. For soft tissues, perform one homogenization cycle at the speed 6 m/s at +/− 2 °C for 30 s.16.**Optional step:** In case a homogenizer instrument is not available, manual tissue disruption can be performed using mortar and pestle with liquid nitrogen.17.Extract the homogenized samples in a shaker for 5 min at RT.18.Centrifuge samples for 10 min at 20,000× *g* at +4 °C.19.Collect the supernatants on a 96-well filter plate and centrifuge for 5 min at 700× *g* at 4 °C.20.**Optional step:** Filter the samples using solvent resistant syringes and PTFE filters into the HPLC vials.21.Take aliquots (5–25 μL) of filtered samples and combine into one vial to be used as QC sample in the analysis.22.Analyze the samples immediately or store the plate at +4 °C maximum of 1 day or −20 °C until analysis.

#### 3.1.2. LC–MS Measurement

The most commonly applied analytical technique in non-targeted metabolic profiling is mass spectrometry, often combined with liquid or gas chromatographic separation at the front end. In order to cover a wide range of polarities among the analyzable metabolites, different chromatographic methods may be utilized, e.g., reversed-phase chromatography (RP) and hydrophilic interaction chromatography (HILIC). MS data can then be acquired in both positive (+) and negative (−) electrospray ionization (ESI) polarities.

23.Use the following conditions for RP chromatography: Column oven temperature 50 °C, flow rate 0.4 mL/min, gradient elution with water (eluent A) and methanol (eluent B) both containing 0.1% (*v/v*) of formic acid. Gradient profile for RP separations: 0–10 min: 2 → 100% B; 10–14.5 min: 100% B; 14.5–14.51 min: 100 → 2% B; 14.51–16.5 min: 2% B. Needle wash with 50% ACN. Set the injection volume at 2 μL and sample tray at 10 °C.24.Use the following conditions for HILIC: Column oven temperature 45 °C, flow rate 0.6 mL/min, gradient elution with 50% *v/v* ACN in water (eluent A) and 90% *v/v* ACN in water (eluent B), both containing 20 mM ammonium formate (pH 3). The gradient profile for HILIC separations: 0–2.5 min: 100% B, 2.5–10 min: 100% B → 0% B; 10–10.01 min: 0% B → 100% B; 10.01–12.5 min: 100% B. Needle wash with 50% ACN. Set the injection volume at 2 μL and sample tray at 10 °C.25.To operate at high mass accuracy (<2 ppm), calibrate the MS daily and use the continuous mass axis calibration by monitoring two reference ions from an infusion solution throughout the analytical runs. Examples of reference ions in ESI+ mode: *m*/*z* 121.050873 and *m/z* 922.009798, and reference ions in ESI− mode *m/z* 112.985587 and *m/z* 966.000725. These reference ions are coming from the compounds in the infusion solution. *m/z* 121 is purine, *m/z* 112 is trifluoroacetic acid and *m/z* 922 and 966 are HP-0921 (Hexakis (1H,1H,3H-tetrafluoropropoxy) phosphazine) [20,21]26.Use the following conditions for Jetstream ESI source: drying gas temperature 325 °C and flow 10 L/min, sheath gas temperature 350 °C with a flow of 11 L/min, nebulizer pressure 45 psi, capillary voltage 3500 V, nozzle voltage 1000 V, fragmentor voltage 100 V and skimmer 45 V. Use nitrogen as the instrument gas.27.For data acquisition, use a 2 GHz extended dynamic range mode in both ESI + and ESI - ionization modes from *m/z* 50 to 1600 (may be adjusted according to sample matrix). Collect the data in the centroid mode at an acquisition rate of 1.67 spectra/s (i.e., 600 ms/spectrum) with an abundance threshold of 150. For automatic data dependent MS/MS analyses, set the precursor isolation width to 1.3 Da. From every precursor scan cycle, 4 most abundant ions are selected for fragmentation. These ions are excluded after two product ion spectra and released again for fragmentation after a 0.25 min hold. Product ion scan time is based on precursor ion intensity, ending at 25,000 counts or after 300 ms. Use collision-induced dissociation voltage 10, 20 and 40 V in subsequent runs.28.Generate the worklist containing analytical samples. Inject quality control samples after every 12 samples and before and after the sample sequence. To monitor contamination during sample preparation and liquid chromatography, inject extraction blanks in the beginning (before the QC samples) and end of the analysis. The injection order of samples should be randomized. If the study contains samples from multiple matrices, such as samples from different organs, it is recommended that all the samples of a matrix be injected consecutively, for example first inject all heart samples, followed by all liver samples. If there are multiple samples from the same individual, it is recommended that the samples of an individual are run consecutively. We use an in-house developed software called Wranglr (github.com/antonvsdata/wranglr) to automate the generation of sample worklists by automatically randomizing the sample order and adding QC and MS/MS samples. Wranglr is an open-source web application developed with the Shiny package for R [22].29.Inject 2 blanks and then 15–20 QC samples at the beginning of each run for column conditioning. Inject a QC sample after every 12 samples during the analysis. At the end of each run, include 4 QC samples: 1 for MS analysis, 3 for MS/MS analysis from 3 different collision energies and finally, 2 blanks. If the run contains samples from different tissues or species (i.e., different expected metabolite profiles), it is recommended to run the MS/MS analysis additionally from one sample per different sample type to increase the coverage of available MS/MS data.

### 3.2. Data Collection and Preprocessing

The data collection (peak picking) and subsequent preprocessing of the raw data are critical steps in non-targeted metabolomics data-analysis since they determine the quality of the data for all the remaining steps (Figure 2). Various peak picking algorithms exist, utilized by vendor-specific and open-source software as well as freely available online services. Widely used examples of open-source software include XCMS (and XCMS Online), MZmine and MS-DIAL. In this workflow, MS-DIAL (http://prime.psc.riken.jp/Metabolomics_Software/MS-DIAL/) [17] is used for the peak picking; it has user-friendly interphase and contains advanced tools for signal filtering, metabolite annotation, chromatogram curation and visualization. After collection of the raw data, pre-processing is required to monitor the quality of the data, make any required transformations/corrections to the data, as well as reduce/merge the number of features originating from the same metabolite.

#### 3.2.1. Peak Picking and Alignment

30.Before the peak picking, convert the raw instrumental data (i.e., *.d) to ABF format using Reifycs Abf Converter (https://www.reifycs.com/AbfConverter). Follow the vendor-specific instructions on the website.31.For the peak picking in MS-DIAL (version 3.70), choose the following parameters:
a.*m/z* tolerance according to the instrument mass accuracy; however, it is advisable to set a bit higher tolerance to avoid screening out peaks close to the threshold, e.g., for QTOF we have used tolerance of 0.01 Da or 10 ppm.b.minimum peak height 2000 signal counts for QTOF (or at least 5 times the typical noise level of the instrument; 3000 signal counts for highly concentrated plant samples).c.mass slice width 0.1 Da (suitable for QTOF and other instruments with high mass accuracy).d.linear weighted moving average as the smoothing method (smoothing level 3 scans and minimum peak width 5 scans, according to developer recommendations).e.in positive mode, select [M + H]^+^, [M + NH_4_]^+^, [M + Na]^+^, [M + K]^+^, [M + CH_3_OH + H]^+^ and [M − NH_3_ + H]^+^ as the most typical adducts and in-source fragments; in negative mode, select [M − H]^−^, [M − H2O − H]^−^, [M + Cl]^−^, [M + HCOOH – H]^−^ and [2M − H]^−^ as the adducts and in-source fragments. Depending on previous knowledge, more adducts may be determined.32.For the peak alignment, set the retention time tolerance according to method accuracy (for the present method we have used 0.05 min and MS1 tolerance at 0.015 Da. Set the detection filter (detected in at least one sample group) at 50%. Unselect the “detected in all QCs” option and select gap filling by compulsion.33.Once the peak picking is finished, export the alignment result as peak areas into a raw data matrix as a tab-separated text file. Transform the data matrix into a datasheet in a spreadsheet software, such as Excel. Insert additional columns to each datasheet specifying the chromatography and the ionization mode before combining the datasheets into a single file. Remove columns containing peak areas from auto-MS/MS data files.

#### 3.2.2. Drift Correction and Flagging Low-Quality Features

LC–MS-based metabolomics suffers from systematic intensity drift during an LC–MS run. This means that the signal intensity of a molecular feature either decreases or increases systematically throughout the experiment. Removing this drift increases the quality of LC–MS data and allows estimating the true biological effects more accurately. Unfortunately, some molecular features show too much variation in the QC intensities even after drift correction. We use here different quality metrics defined by Broadhurst et al. [10] for measuring the quality of a molecular feature before and after drift correction. Low-quality features are flagged and not included in downstream data analysis. Note that we do not recommend removing low-quality features completely, as they are sometimes needed in the metabolite identification phase when searching for specific ions or fragments of known molecules.

34.Make sure that missing values are correctly represented. A peak picking software might use a numerical value (such as 0, 1 or -999) to represent missing values, while other software such as R have specific ways of representing missing values. For more information on handling missing values, see Section 3.2.4.35.Molecular features with too low detection rate in the QC samples should be flagged. We recommend a threshold is 70%, meaning that a molecular feature needs to be detected in at least 70% of the QC samples.36.Log-transform the features prior to drift correction. Log-transformed data normally conform better with the assumptions of the regression model used to model the drift. We use the natural logarithm. Replace zeroes with a value slightly above one (e.g., 1.1) to make sure that all log-transformed values are > 0.37.The drift correction should then be performed by repeating steps 38–40 for each molecular feature. These procedures are included in notame (function correct_drift()).38.Model the drift function (*f_drift_*) by fitting a smoothed cubic spline [23] to the QC samples, where the abundance of the molecular feature is predicted by the injection order Figure 3a. Smoothed cubic spline regression has one hyperparameter: a smoothing parameter, which controls the overall curvature of the drift function. The smoothing prevents the spline from overfitting the drift function in the presence of a few deviating QC samples (see Figure 4). A suitable value for the smoothing parameter is chosen by leave-one-out cross validation. For the R function smooth.spline, [24] we recommend the smoothing parameter to be between 0.5 and 1.5.39.Correct the abundance of each sample using the following formula (for a sample with injection order *i*):
(1)$$x_{corrected}\left(i\right)=x_{original}\left(i\right)+mean\left(x_{QC}\right)-f_{drift}\left(i\right)$$40.Reverse the log transformation by applying the corresponding exponential function.41.The drift correction procedure is visualized (Figure 3 and Figure 4) by drawing a scatter plot of the abundances against the injection order before and after drift correction. A line representing the drift function should be added to the scatter plot before correction. To reduce the amount of manual inspection, we usually only inspect potential candidate molecular features selected from downstream statistical tests.42.**Optional step:** Compute the quality metrics after drift correction and keep only the drift-corrected values for the molecular features where the change in quality metrics indicate that the data quality has been improved. For the other molecular features, retain the original values.43.Flag or remove low-quality features. As recommended by Broadhurst et al. [10], only the molecular features with RSD < 0.2 and D-ratio < 0.4 should be retained. In notame, this can be done with the function flag_quality().

#### 3.2.3. Quality Control

The raw data obtained from the peak picking software requires careful examination to estimate the need for additional preprocessing such as drift correction (see 3.2.2.). In the now proposed workflow, the data quality is monitored at each step of the preprocessing with a set of visualizations. Example figures are based on RP positive data from a dietary intervention study [25], before and after drift correction and removal of low-quality features. All the visualizations described in this section are available in notame (see the visualizations vignette for details).

44.Draw the visualizations in steps 46-52 before drift and after drift correction.45.Flag low-quality features to monitor data quality and the effect of preprocessing.46.Apply a linear model to each feature, where the feature levels are predicted by injection order. Fit the model separately for QC samples, biological samples and all samples. Then visualize the effect of drift correction to individual features by drawing histograms of the *p*-values for the regression coefficient of injection order (Figure 5). We represent the expected uniform distribution by a horizontal line. Ideally, the *p*-values should roughly follow the expected uniform distribution, which would mean that there is no systematic dependency between feature abundances and injection order [26]. Unfortunately, this is rarely the case, but the closer the distribution is to uniform, the better. It is recommended to apply this procedure separately on QC samples and biological samples, which allows observing the drift patterns in both parts of the dataset.47.Draw boxplots (Figure 6) where each individual boxplot represents the distribution of all feature levels in a sample: in the first boxplot order the samples by study group (a.1, a.2) (and possibly time point). This can reveal systematic changes in the global feature levels across samples. In the second type (b.1, b.2) order the samples by injection order, highlighting the QC samples. This allows us to observe any systematic drift across the feature levels in the samples.48.Before subsequent visualizations, mean center the features and divide by standard deviation.49.Visualize the distribution of the Euclidean distances between samples using a density plot. The plot should feature two distributions, the distribution of distances between QC samples and the distances between biological samples. Ideally, the distribution of QC sample distances should be narrow and well separated from the distribution of study samples (Figure 7).

Principal component analysis (PCA) [27,28,29] or t-distributed stochastic neighbor embedding (t-SNE) [30] can be used for observing patterns in the data by drawing scatter plots of the samples in a low-dimensional space (Figure 8 and Figure 9). PCA is a linear method, while t-SNE can also reveal non-linear patterns. Unlike t-SNE, PCA offers information on loadings, i.e., on how the principal components are constructed from original features. For these reasons, we consider PCA and t-SNE as complementary methods. For conciseness we only show t-SNE figures here.

50.Draw scatterplots of the data points using PCA and t-SNE. Samples can be highlighted by coloring the points in the scatter plot with a study factor (e.g., treatment groups or time points) to observe trends in the data. Ideally, QC samples should cluster together (Figure 8). We also draw separate plots where the samples are colored by injection order to observe drift patterns (Figure 9). If the data quality is high, there should be no visible patterns according to injection order (Figure 9b).51.**Optional step:** If there is a large number of samples and the points in the t-SNE plots tend to overlap, draw a hexbin version of t-SNE scatter plots colored by injection order (Figure 10), where the plot area is divided into hexagons and each hexagon is colored by the mean of the injection orders of the points inside that hexagon. As before, in an ideal case, there should be no visible drift patterns.52.Apply hierarchical clustering [31,32] to the samples and visualize the result in a dendrogram (Figure 11a,b). The QC samples should cluster together early. We also draw a heatmap (Figure 11c,d) representing pairwise distances between samples, where samples on the x and y axes are ordered by hierarchical clustering. The QC samples should have smaller inter-sample distances than other samples. Several techniques can be used for clustering. However, we have consistently achieved good results with hierarchical clustering using Euclidean distances and Ward’s criterion for linking clusters [32].

#### 3.2.4. Imputation, Transformation, Normalization and Scaling

Missing data occur in metabolomics datasets for various reasons and managing this missingness is highly challenging [33]. Imputation is the procedure of replacing missing data with reasonable values using a priori knowledge or information available from the existing data. In this workflow, we perform random forest (RF)-based imputation using the missForest package [33,34], although several other procedures are available [35,36]. Data distributions can affect statistical analysis, especially for variance-based models [37]. Consequently, transformation and normalization can be used to adjust for data heteroscedasticity and skewed distributions among the molecular features. Depending on the type of multivariate analysis chosen we will proceed with different normalization and transformation approaches [38], however in the case of the feature-wise univariate analysis (Section 3.3.1) only imputation is performed. All the preprocessing methods mentioned here are provided in notame (see the preprocessing vignette for details).

53.Impute missing values using random forest imputation. QC samples should be removed prior to imputation to ensure that the imputation is based on patterns in the biological data.54.Transform the data using either natural logarithmic (nlog) or the generalized logarithmic (glog) function when the data are heavily skewed [38].55.Normalize the data by probabilistic quotient normalization (PQN) [38,39].56.Perform mean centering and scaling by standard deviation (autoscaling), before multivariate analysis; this is necessary with GLM-based methods as well as PCA and PLS-DA. However, this is not required for scale invariant techniques such as RF [40].

#### 3.2.5. Clustering Molecular Features Originating from Same Metabolite

Now used peak picking software can detect isotopes, most common adducts and some in-source fragments and combine those features into one entry in the data matrix. However, in LC–MS analysis, unpredictable adduct behavior and neutral loss formation occurs frequently, resulting in the same metabolite being redundantly represented in the data matrix, causing problems not only for the identification of the compounds but also potentially in the data-analysis step due to multiple collinearities.

We present here a method for clustering and combining these features. This approach was developed bespoke to our workflow [41]. Partially similar methods to tackle this problem have been published also elsewhere [42,43,44]. Features originating from the same compound are assumed to be strongly correlated and have a small difference in their retention time. Thus, the algorithm initially identifies pairs of correlated features within a specified retention time window. The user specifies both the correlation threshold and the size of the retention time window. For illustration, a correlation coefficient threshold of 0.9 and a retention time window of ±1 s are used. Spearman’s correlation coefficient is used, as the relationship between features originating from the same compound is assumed linear. However, this assumption may not hold true if some measured features are close to lower or upper limit of quantification (LLOQ and ULOQ) of the instrument.

Next, an undirected graph of all the connections between the features is generated, where each node represents a feature and each edge represents the corresponding correlation coefficient under the retention time constraint (Figure 12a). The algorithm recursively identifies clusters presumed to reflect the same analyte. In brief, this is achieved using a connectivity criterion, i.e., that the features within a cluster should have strong correlation to a sufficient number of the other features within the cluster. A detailed explanation of the algorithm is beyond the scope of this paper and has been included in the Appendix A (Section 1: Clustering features originating from the same compound) for more advanced (bio) computational scientists.

After clustering, the feature with the largest median peak area per cluster is retained. All the features that are clustered together are recorded for future reference. Figure 12b shows the state of the graph from Figure 12a after clustering, with each final cluster colored differently.

57.Cluster the molecular features from each analytical mode separately using the algorithm described above. Represent each cluster with the feature with the highest median abundance. Use these features for multivariate analysis and the clustering information for metabolite identification. The algorithm is provided in notame through the cluster_features() function.

### 3.3. Data Analysis

Once the raw data are checked for quality and analytical drift and the features originating from same metabolites merged to reduce the data matrix, the next phase is to utilize data analytical methods to discover the metabolites of biological importance within the taken study set-up. Preferably, a combination of feature-wise and multivariate analyses can be applied (Figure 2). Notame provides an interface for all the statistical tools mentioned in this section (see the statistics vignette for details).

58.Combine the features from the different analytical modes to a single data matrix. In notame, this is achieved with the function merge_metabosets, which simply concatenates the data matrices and feature information tables row-wise (each row corresponds to a feature) and preserves the sample information unchanged. Note that combining analytical modes inevitably results in increased redundancy in the data matrix, as many compounds are detected in multiple analytical modes. However, combining the analytical modes is necessary so that all available information is available for multivariate analysis methods.

#### 3.3.1. Feature-Wise (Univariate) Analysis

In feature-wise analysis, two types of testing may be used depending on the data: parametric and non-parametric test [45]. The choice of the test statistical depends on the data and the biological questions of the study. Most typically parametric tests are used, but if the features do not satisfy the assumptions of parametric tests, they may be replaced with non-parametric alternatives. Non-parametric methods perform better when dealing with non-normal populations, unequal variances and unequal small sample sizes.

59.For study designs with two groups and no covariates, such as case versus control studies, use a simple Welch’s t-test, i.e., the extension of Student’s t-test to manage unequal variances between groups. For a non-parametric alternative, consider a Mann-Whitney U test.60.For studies with multiple groups, first apply Welch’s one-way analysis of variance (ANOVA), which can manage unequal variances between groups, to select interesting features based on overall *p*-value. To investigate differences between groups, conduct post-hoc pairwise Welch’s t-tests.61.For studies with two categorical study factors, apply two-way ANOVA, which allows examining the main effect of each factor and their interaction. If one or both factors have multiple levels, select interesting features based on overall *p*-values and conduct post-hoc pairwise t-tests as above (bullet 59). For a non-parametric alternative, consider Friedman test.62.For studies with repeated measurements, use a linear mixed effects model with the time point, group and their interaction factors as fixed effects and the subjects as a random effect. If there are no more than two groups or time points, use t-tests on the regression coefficients to assess the significance of the effects. In the case of multiple groups and/or time points, use type III F-tests for ANOVA-like tables, e.g., with the help of the R packages lme4 and lmerTest that provide all the necessary tests [46,47].63.To test the strength of association between molecular features or between molecular features and other variables, use Pearson correlation or Spearman correlation as a non-parametric alternative. This can also be done post-hoc, after identification of key metabolites [14].64.After performing feature-wise tests, *p*-values should be adjusted for multiple testing. We recommend using the Benjamini–Hochberg false discovery rate (FDR) approach. Note that FDR-adjusted *p*-values are frequently referred to as q-values. [45,48,49].

#### 3.3.2. Multivariate Analysis

There are several powerful multivariate tools for analysis of metabolomics data. Dimensionality reduction methods like PCA or t-SNE enable us to explore the data to identify outliers and patterns among samples. Unsupervised clustering methods, such as hierarchical clustering are useful for validating findings from dimensionality reduction methods, as they allow us to observe clustering patterns in high-dimensional space.

Supervised learning techniques, such as partial least squares (PLS) and random forest (RF) are useful for identifying the most interesting molecular features [50,51]. Both the PLS and RF algorithms can be used for both regression and classification purposes. In the case of classification, the PLS model is normally referred to as partial least squares discriminant analysis (PLS-DA). Contrary to the unsupervised methods, supervised methods rely on known outcome or response (e.g., class membership) of each sample and can be used for predictive and descriptive modeling as well as for discriminative variable selection. RF is highly flexible with 3 main advantages over PLS: RF does not assume Gaussian distribution of the variables; RF does not assume linear relationships between response and (latent) predictor variables; Finally, RF is scale invariant, which circumvents issues with scaling and transformations of metabolomics data. On the other hand, it should be noted that PLS can produce stronger models if model assumptions are met. Both PLS and RF offer statistics for evaluating the importance of individual features, such as the variable importance in projection (VIP) values in PLS and Gini index or mean increased error in RF.

65.Apply multivariate algorithms for prediction and variable selection. We employ the MUVR package in R which includes both RF and PLS [50]. For each analysis, three different models are obtained: the minimal-optimal (‘min’), ‘mid’ and all-relevant (‘max’) models (Figure 13). The ‘max’ model corresponds to maximum information content once the non-informative features have been removed and includes the highest numbers of relevant molecular features, thought it may include some redundant features or highly correlated features. This model is normally selected when e.g., pathway analysis will be applied afterwards. The ‘min’ model corresponds to the minimal-optimal set of molecular features where the strongest biomarker candidates are likely to be found. The ‘mid’ model corresponds to a compromise (geometric mean) between the ‘min’ and ‘max’ options, representing and with some redundancy between molecular features. In the end, the selection of the model depends on the research interest and study question, such as pathway analysis (‘max’), best prediction (‘mid’) or biomarker discovery (‘min’).66.**Optional Step**: Follow this step if the MUVR package is not available (for example if other software than R is used). Evaluate performance of the multivariate model. Use cross-validation for PLS and out-of-bag error estimate for RF (for more information see [51])If the model performance is satisfactory, record variable importance metric (VIP value for PLS and rise in error rate for RF) for each feature.

#### 3.3.3. Ranking and Filtering for Variable Selection

After the completion of both feature-wise and multivariate analysis, results are combined via a ranking method in order to determine the most robust and presumably biologically relevant metabolic features to undergo identification.

67.The first step is to sort the molecular features according to their ranks that received though the variable selection process, with the lowest rank or the most important rank (depending on the software) being the 1st rank and the biggest rank or the least important rank being the nth rank (n here is equal to the total number of molecular features available from the variable selection method). In the MUVR package, the output from the ‘min’, ’mid’ or’ max’ models provides the ranks for each of the molecular features already sorted by the smallest rank. The smallest rank represents that this particular molecular feature is the most important one.68.Similarly, for each univariate model, the molecular features are sorted based on their q. The 1st rank is given for the feature with the lowest q-values from the FDR correction and the nth rank for the largest one.69.Then, the rank from the RF model e.g., ‘mid’ model for each molecular feature is added together with the rank from the same molecular feature for the feature-wise model creating a new column with the Final Ranks.70.The choice of the total number of the molecular features that are selected in the end for further analysis e.g., identification or pathway analysis is dependent strictly on the user.71.**Optional Step**: In case the MUVR package is not used for variable selection, the procedure of ranking the molecular features stays the same for any type variable selection is chosen.

### 3.4. Visualization of Results

After feature-wise and multivariate analysis, we recommend visualization of patterns of the dataset, both on a feature level and a global level as well as visualization of the *p*-values and effect size measures, to offer a broad view of the results. All the visualizations in this section are provided in notame unless stated otherwise (see the visualizations vignette for details).

#### 3.4.1. Feature-Wise Graphs

While t-SNE figures (Figure 8 and Figure 9) provide a solid overview of the overall patterns in the data, visualizing effects of study factors on a molecular feature level is useful when interpreting the results. The visualization type used depends on study design.

72.If the study has multiple study groups, the differences between groups can be illustrated by beeswarm boxplots separately for each group (Figure 14).73.If the study contains samples from multiple time points, the effect of time can be visualized with a line plot using one line per subject together with a thicker line representing the mean at every time point (Figure 15).

If the study contains both multiple groups and multiple time points, consider the following visualizations:

For repeated measures data, plot least square means from the repeated measures model for each study group. You should also add whiskers around the points representing 95% confidence intervals, standard deviation or other measure of variability (Figure 16).

74.Draw a line plot similar to the one in step 73, but color the subject lines according to group and draw separate mean lines for each group (Figure 17a). If the figure gets too cluttered, consider plotting each group separately in small multiples, with a common y-axis (Figure 17b).

#### 3.4.2. Comprehensive Visualization of Results

Here, we present ways of visualizing results from both feature-wise and multivariate analysis. For illustration, we use a simple case from the RP positive mode of an intervention study, where the samples were taken from two time points, before and after an intervention. For feature-wise analysis, we used a linear model with individual molecular feature as the dependent variable and the time point as the independent variable. We also calculated fold change between the two time points for a scale-free measure of effect size. For multivariate analysis we fit a PLS-DA model predicting the time point from the features.

75.Visualize the patterns in the final dataset using unsupervised dimensionality reduction techniques such as PCA [28] (Figure 18) and t-SNE. If the PCA score plot reveals interesting patterns, use a PCA loadings plot to reveal which features contribute the most to the visualized components.76.If PLS(-DA) is used, visualize the samples in a PLS score plot (see Figure 19).77.To visualize overall changes with respect to time in studies with multiple time points, use PCA and t-SNE figures with arrows depicting change in each individual. The arrows should start at the first time point and end at the last time point for each individual. We recommend plotting each study group separately, as the plot can get crowded since the arrows occupy significantly more space than points (Figure 20).78.Visualize the distribution of *p*-values from feature-wise analysis in a histogram. Use a line to depict the expected uniform distribution (under null hypothesis). If the distribution of the *p*-values deviates from the line as in Figure 21, it can be argued that we are observing a real effect.79.Visualize the results of feature-wise tests in a volcano plot. Volcano plots are scatter plots with *p*-values on the y axis and effect size (such as fold change) on the x-axis. Add a horizontal line representing the significance threshold for FDR-adjusted q-values. To co-visualize multivariate results, the features can be colored by their relevance score in the multivariate prediction (Figure 22).

Manhattan plots are commonly used in genome-wide association studies (GWAS) to visualize the location of the most significant single nucleotide polymorphisms on the genome. Manhattan plots can be applied in metabolomics by using mass-to-charge ratio or retention time on the x-axis. In addition, in cases where direction of effect can be determined, we can multiply the y-axis by the sign of the effect to create so-called directed Manhattan plots. The Manhattan analogy is not lost since the downward points represent the reflection of the skyline on the Hudson River. Note that Manhattan plots should always be drawn separately for each column and ionization mode, as the metabolite classes corresponding to certain *m/z* and retention time values depend on the column and ionization mode used.

80.Use a Manhattan plot to connect the results of statistics to biochemical properties of the molecular features. The Manhattan plot should have either retention time or mass-to-charge ratio as the x-axis and –log10(*p*-value) on the y-axis. For a directed Manhattan plot, multiply –log10(*p*-value) by the sign of the effect. The points in the Manhattan plot can be colored by the respective VIP value from PLS-DA or another similar metric. Similar to volcano plots, add a horizontal line to represent the significance threshold for FDR-adjusted q-values. Figure 23a,b show Manhattan plots with mass-to-charge ratio and retention time on the x-axis, respectively.81.To combine the information of both Manhattan plots, consider a scatter plot with *m/z* and retention time on the x- and y-axis, with the size of the point reflecting *p*-value and potentially colored by variable importance from multivariate modelling (e.g., VIP; Figure 24) or by effect size (e.g., fold change; not shown). While size is not an accurate metric in visualizations, this visualization combines mass and retention time so that the most interesting metabolite classes can be identified. As with Manhattan plots, these plots should be drawn separately for each column and ionization mode.

We utilize Multiple Experiment Viewer (http://mev.tm4.org/) for *k*-means clustering and hierarchical clustering analyses, which group metabolites into separate clusters or into a hierarchy tree, respectively. Multiple Experiment Viewer is a useful option for post-hoc analysis as it requires no programming expertise. Readers familiar with programming can use other tools for similar results.

The heat maps produced from the analyses can be used to assess the impact of the intervention and the number and proportion of metabolites behaving in a certain manner (Figure 25). We also use the notame R package to produce heat maps of the identified metabolites and their associations with e.g., clinical markers, in which case additional information may be added to each cell, such as the statistical significance with circles, where a larger circle represents a lower *p*-value.

82.For the clustering in Multiple Experiment Viewer, first normalize the rows (signal abundances) and select appropriate color scale limits for the normalized abundances (0 to 10% of features can be off limits). For hierarchical clustering, choose whether to cluster only the features or samples as well. Use Pearson correlation and average linkage clustering. For k-means clustering, choose cluster genes, use Pearson correlation, calculate k-means and choose a low number of clusters (e.g., 4) for the initial run. Repeat the procedure by increasing the number of clusters until no more clusters with a unique pattern emerge and choose the highest number of clusters based on this visual optimization.

### 3.5. Identification of Metabolites

The identification and annotation of metabolites is a critical step in any metabolomics study to attribute biological meaning to the data analytical results and to enable further hypotheses to be developed for subsequent studies. In recent years, the development of new software and online tools as well as the emergence and expansion of publicly available spectral databases of metabolites have greatly facilitated the identification process [52,53]. Nevertheless, metabolite identification remains perhaps the most time-consuming task where manual curation is necessary and where not all detected molecular features can be identified, leaving knowledge gaps for the interpretation of the results. Alongside with the challenges related to the instrumental differences and matching the obtained MS/MS data to databases, a key bottleneck restricting the level and number of identifications is the lack of reference data for the vast number of metabolites produced by living organisms, estimated up to one million for the plant kingdom [54] and more than 40,000 for humans [55]. Likewise, matching the obtained MS/MS data to existing databases is not straightforward due to differences in experimental conditions used for collecting the reference data. Other limitations may be related to poor quality or lack of mass spectra from metabolites with low abundance in the sample.

We utilize MS-DIAL [18] in the initial semi-automated step of metabolite identification, where the experimental characteristics (exact *m/z*, retention time where applicable and MS/MS spectra in CID voltages 10, 20 and 40 V) are compared with those in databases available in NIST MSP format. These databases include MassBank [53], MoNA [56] and others available from the RIKEN Center for Sustainable Resource Science website (http://prime.psc.riken.jp/Metabolomics_Software/) combined in single files for the positive and negative ionization mode. Additionally, we have included our in-house spectral library in the MSP files. The semi-automated identification process annotates metabolites with similarity score 80% or above, after which the annotations are manually curated by assessing the similarity of the MS/MS spectra and the alternative annotations proposed by the software.

After the curation of the metabolites annotated by MS-DIAL, the remaining unknown metabolites undergo additional searches in databases that are primarily available online, including METLIN [52] for small metabolites and LIPID MAPS [57] for unknown metabolites with RP retention time in the lipid region (> 9 min). Additional attempts to characterize the unknowns are made utilizing MS-FINDER [18], which 1) calculates and scores the possible molecular formulas based on the exact mass and isotopic pattern, 2) searches for compounds corresponding to the likely molecular formulas from non-spectral chemical libraries and 3) compares the experimental MS/MS spectrum of the unknowns with *in silico*-generated MS/MS spectra of the candidate structures.

#### 3.5.1. Comparison with Pure Standard Compounds (MSI Level 1)

83.For the identification of metabolites (identification level 1 according to the Metabolomics Standards Initiative) [58], compare the molecular features against an in-house library (i.e., a reference standard analyzed previously with the same platform in the same chromatographic conditions). Apply the following criteria:a.matching *m/z* (within 10 ppm or according to instrument mass accuracy);b.similar retention time (ΔRT < 0.2–0.5 min), taking into consideration any possible near-eluting isomers.c.MS/MS spectra (main fragments matching within 0.02 Da in one or more CID voltage)

#### 3.5.2. MS/MS Fragmentation and Database Comparison (MSI levels 2–3)

84.For the putative annotation of metabolites (ID level 2), compare the mol features against publicly available spectral databases, including a database file (compiled in MSP format for using within MS-DIAL) and online databases. The annotation has acceptable reliability if the main fragments (excluding the molecular ion) match between the experimental and reference MS/MS spectra in only one proposed metabolite. In case several alternatives exist with similar MS/MS, the common denominator of all the alternatives (e.g., a compound class, ID level 3) is given as the annotation instead. Apply the following criteria:
a.matching *m/z* (within 10 ppm or according to instrument mass accuracy)b.MS/MS spectra (main fragments matching within 0.02 Da)85.For the putative characterization of compound class (ID level 3), use the following approaches to obtain characteristic information of the metabolite:
a.Compare the experimental MS/MS with in-silico generated spectra in MS-FINDER;b.Use the calculated molecular formula, retention time and diagnostic MS/MS fragments to determine the compound class.

#### 3.5.3. Pathway Analysis

Once molecular features are annotated as metabolites, pathway analysis may be conducted to better understand the biological relevance of the metabolites, as well as their involvement in metabolic pathways, e.g., related to intervention effects of disease etiology [1,3]. We consider identification of metabolites until level 2 (putative annotation) to be essential prior to pathway analysis. Of the several pathway analysis tools that are freely available, we use predominantly MetaboAnalyst and Cytoscape. For both tools, conversion of metabolite name to HMDB or KEGG ID that are generally recognizable by the pathway analysis software is essential, since one molecule can have multiple names according to the preference of each research group.

86.Option 1: In MetaboAnalyst [59] (https://www.metaboanalyst.ca/) use Enrichment or Pathway Analysis which enables enrichment and visualization of metabolic pathways in which the metabolites could potentially be involved. For more detailed information about metabolic regulation, the Network Explorer enables inclusion of fold change data, along with gene expression data.87.Option 2: Cytoscape [60] (https://cytoscape.org/) is a powerful stand-alone tool that is used by biomedical researchers to visualize and dynamically analyze gene/protein/metabolite interaction networks. The strength of Cytoscape is even more apparent when linked to databases, e.g., MetScape [61], which allows for visualizing and interpreting metabolomic data in the context of human metabolic networks.

A step-by-step instruction to use the software is listed in the Appendix A (Section 2: Tutorial on Pathway Analyses Tools). It is worth to mention that pathway analysis may not be helpful for lipids, due to i) the limitation of the non-targeted LC–MS metabolomics platform to differentiate the position of the double bonds within the lipid molecule, which impairs the translation of lipid identity to KEGG or HMDB ID and; ii) that most pathway analysis tools would group certain lipid classes that vary greatly based on their fatty acid composition to one node, which may not be biologically meaningful. As an example, phosphatidylcholines with different acyl composition, will be grouped into one node of phosphatidylcholine regardless of the acyl composition, which may not accurately represent acyl transfer in vivo. This gap hence emphasizes the need of pathway analysis tool specialized for lipid molecules.

### 3.6. Biological Interpretation of the Results

The analytical procedure described above is aimed to identify metabolites and metabolic pathways that are affected in the chosen study design e.g., differences in circulating metabolites after dietary or other interventions or processing-induced alterations to the phytochemical composition of a certain food. While the described workflow is efficient in elucidating such metabolites, the ultimate value lies in the demonstration of biological significance. The findings need to be related to the scientific context and interpreted in the light of existing biological knowledge. Optimally, findings can be validated e.g., in subsequent studies, where the most interesting/important metabolite species may be chosen for additional analysis, often encompassing development of targeted, quantitative analytical approaches and analyzed in different study populations. An example of such approach is the recent discovery of various trimethylated compounds related to whole grain consumption [62] and the establishment of a quantitative method within another cohort [63].

## 4. Conclusions

Non-targeted metabolic profiling analysis employing liquid chromatography and mass spectrometry analysis has proven its usefulness in various fields of natural and medical sciences during the last couple of decades and has greatly improved our capabilities to explore and understand the chemical space in biological samples. Notame workflow encompasses all the essential steps in metabolic profiling studies, from generation of samples to the interpretation of the results and is aimed to serve as a general guideline for setting up and executing metabolomics studies, as well as support users with an in-housed developed R package (notame, version 0.0.1 https://github.com/antonvsdata/notame).

## Figures and Tables

**Figure 1 metabolites-10-00135-f001:**
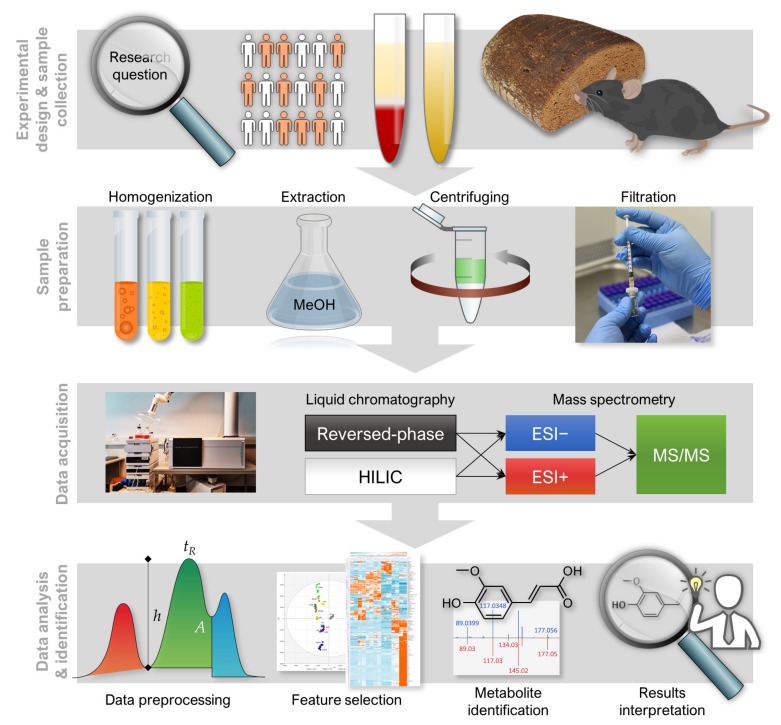
A general overview of notame workflow containing four important stages; 1. Experimental designs and sample collection, 2. sample preparation, 3. data acquisition, 4. data analysis and biomarker identification analysis.

**Figure 2 metabolites-10-00135-f002:**
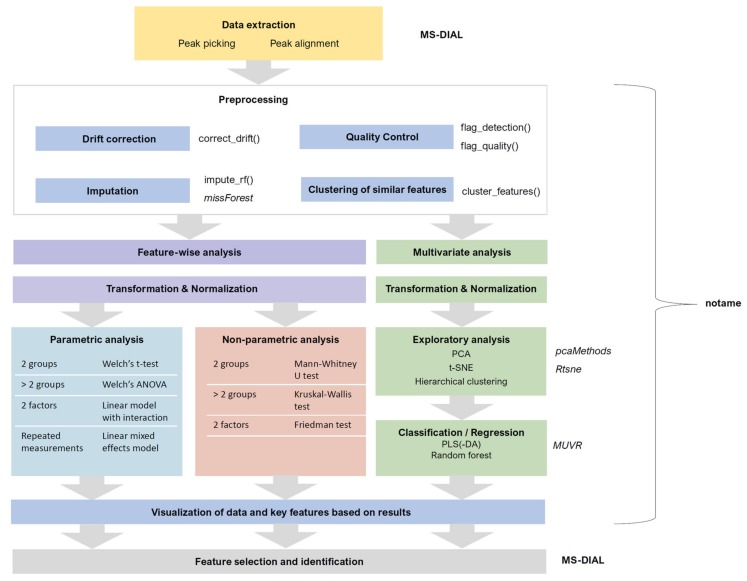
Workflow of the statistical analysis after the peak-picking step. The choices depend on the type of data, the research question and the study design. The tools used for specific steps are listed on the right side of the respective steps. Italicized names are names of external R packages, names ending with () are major functions from the notame package. For more details, see the package documentation.

**Figure 3 metabolites-10-00135-f003:**
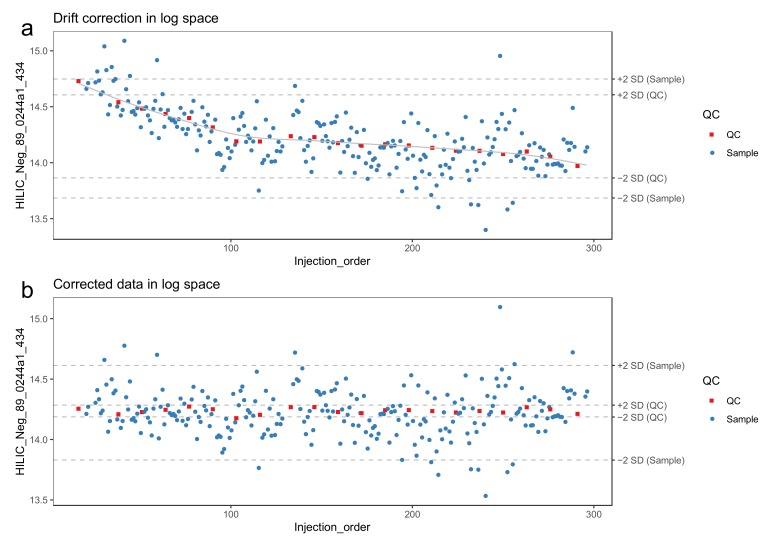
A molecular feature before (**a**) and after (**b**) drift correction by smoothed cubic spline regression. The horizontal lines represent 2 standard deviations from the mean of quality control (QC) samples and biological samples, respectively. The systematic effect of the drift is reduced upon correction.

**Figure 4 metabolites-10-00135-f004:**
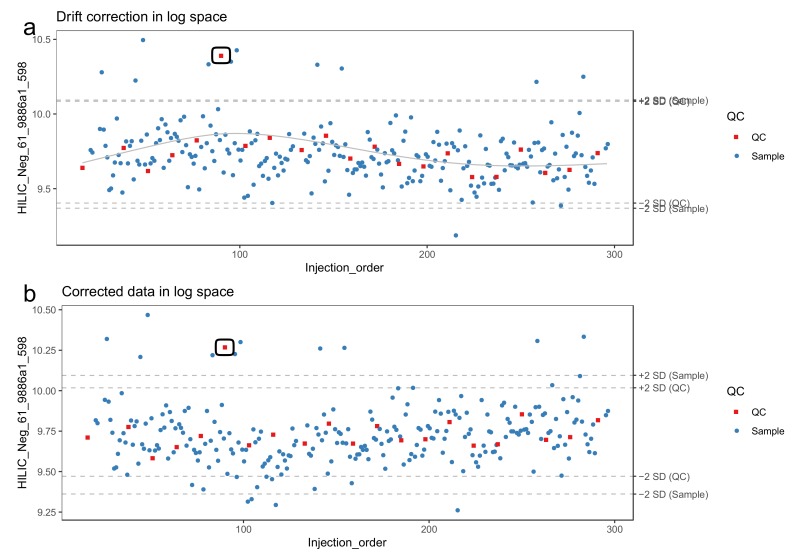
A molecular feature in the presence of an outlying quality control (QC) sample (circled) before (**a**) and after (**b**) drift correction by smoothed cubic spline regression. The horizontal lines represent 2 standard deviations from the mean of QC samples and biological samples, respectively. Due to the smoothing, the correction method is robust against the deviating QC sample and adjusts seemingly adequately for the global drift trend.

**Figure 5 metabolites-10-00135-f005:**
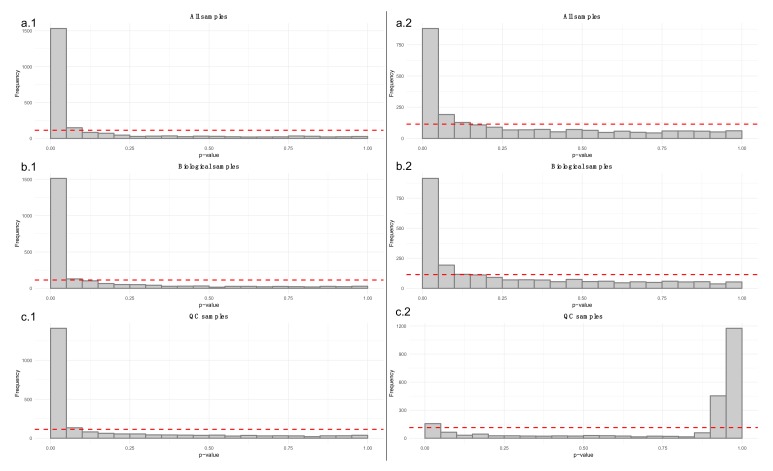
The six histograms illustrate *p*-values from linear regression models between each feature and injection order. The dashed red lines represent the uniform distribution. The a.1 and a.2 histograms show the *p*-values from before (**a.1**) and after drift correction (**a.2**) in all the samples. The b.1 and b.2 histograms focus only in the biological samples before (**b.1**) and after (**b.2**) drift correction. Finally, the c.1 and c.2 histograms show only the *p*-values from the quality control (QC) samples before and after drift correction. In this case, we have a strong drift in the LC–MS data because the *p*-values of the QCs (**c.1**) tend to gather close to zero. After the drift correction, (**c.2**), *p*-values for the QCs are increased.

**Figure 6 metabolites-10-00135-f006:**
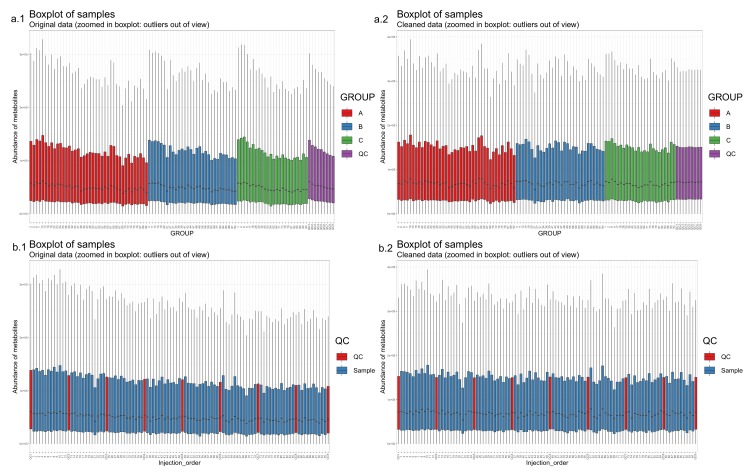
Boxplots of feature intensities per sample. The boxplots (**a.1**), where the samples are ordered by study group (**a.1**) and (**b.1**), where the samples are ordered by injection order and quality control (QC) samples are colored distinctly (**b.1**), show a clear systematic decrease in signal intensity during the injection sequence. After the drift correction, the drift pattern is no longer observable (in boxplots **a.2** and **b.2**).

**Figure 7 metabolites-10-00135-f007:**
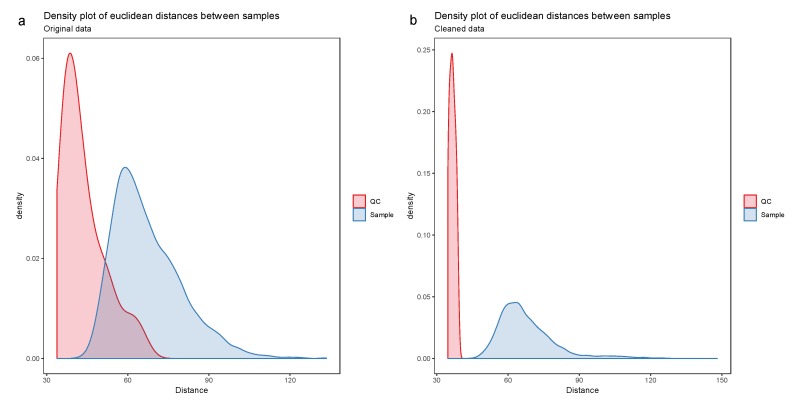
The density plot (**a**) shows a clear overlap between the distribution of quality control (QC) samples and the biological samples, which indicates poor data quality. After drift correction and quality control (**b**), the distributions are no longer overlapping.

**Figure 8 metabolites-10-00135-f008:**
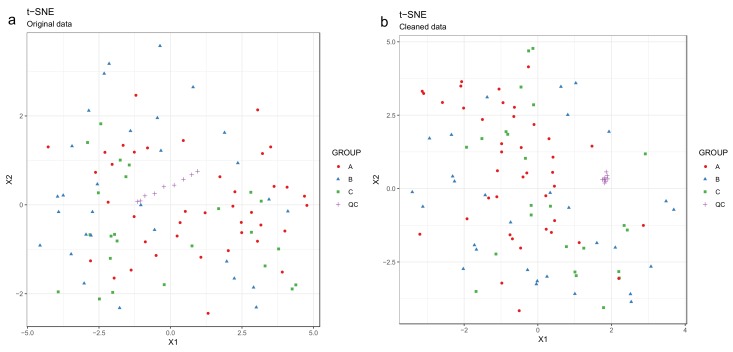
Investigating drift correction patterns using the t-SNE method. The quality control (QC) samples are shifting systematically before drift correction (the line trend of the purple crosses symbol) (**a**), whereas after the drift correction (**b**), the line trend of the QCs is gone and the QCs are now group nicely.

**Figure 9 metabolites-10-00135-f009:**
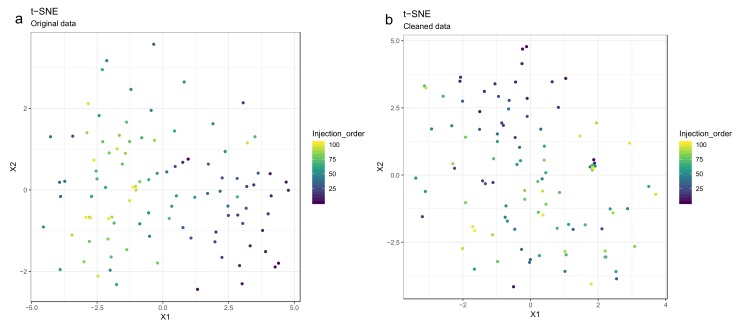
The drift pattern in the injection order (the color trend) using the t-distributed stochastic neighbor embedding (t-SNE) method is visible before drift correction (**a**), whereas after drift correction (**b**), the samples are more randomly scattered.

**Figure 10 metabolites-10-00135-f010:**
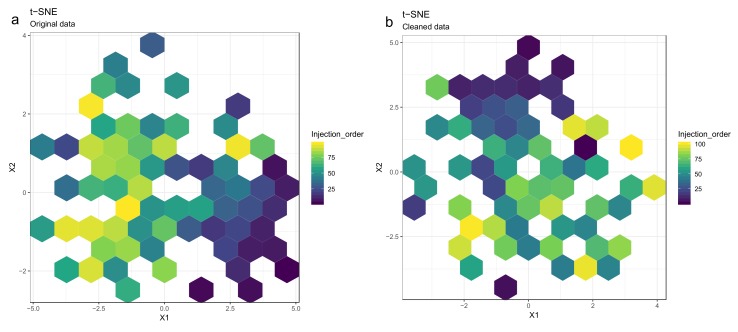
The hexbin plots show similar patterns as the scatterplots in Figure 9: The drift pattern in the injection order (the color trend) using the t-distributed stochastic neighbor embedding (t-SNE) method is visible before drift correction (**a**), whereas after drift correction (**b**), the samples are more randomly scattered. The color of each hexagon corresponds to the mean injection order of the data points in that hexagon.

**Figure 11 metabolites-10-00135-f011:**
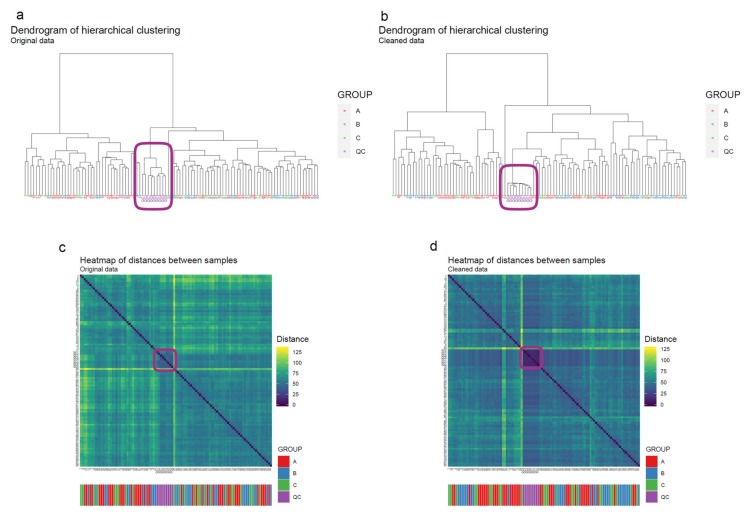
The hierarchical clustering algorithm clusters quality control (QC) samples together even before drift correction (**a**) whereas, after performing drift correction (**b**), the QC samples cluster more clearly together. In the heatmap after the drift correction (**d**) a QC “block” pattern (purple color code), is more clearly visible than in the heatmap before drift correction (**c**).

**Figure 12 metabolites-10-00135-f012:**
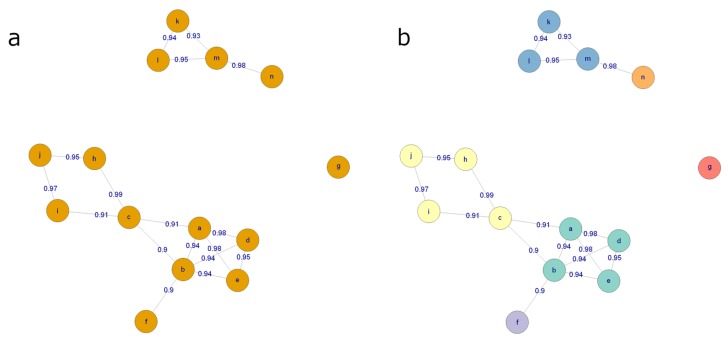
(**a**) An example graph, where every node is a molecular feature and every edge represents a high correlation coefficient and a small retention time difference between the features. (**b**) The graph after the clustering procedure. Each color corresponds to a distinct cluster of features.

**Figure 13 metabolites-10-00135-f013:**
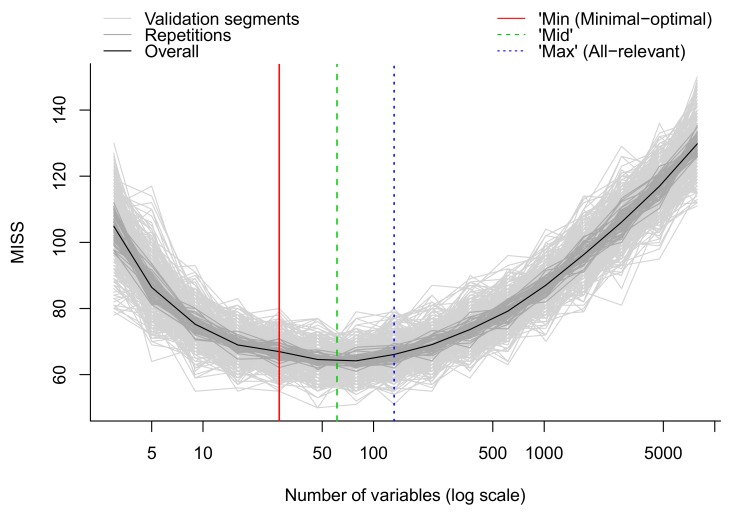
Modelling error measured as the number of miss-classification during internal cross-validation in MUVR. The overall modelling error (black curve) initially decreases when removing noisy variables until the ‘max’ model. Further removal of variables until the ‘min’ model removes redundant features while keeping modeling error almost constant. The ‘mid’ model represents a compromise between the ‘min’ and ‘max’ models and a theoretical optimum model. Light and dark grey lines represent higher level of detail in the validation procedure and we refer to Shi et al. [50] for details.

**Figure 14 metabolites-10-00135-f014:**
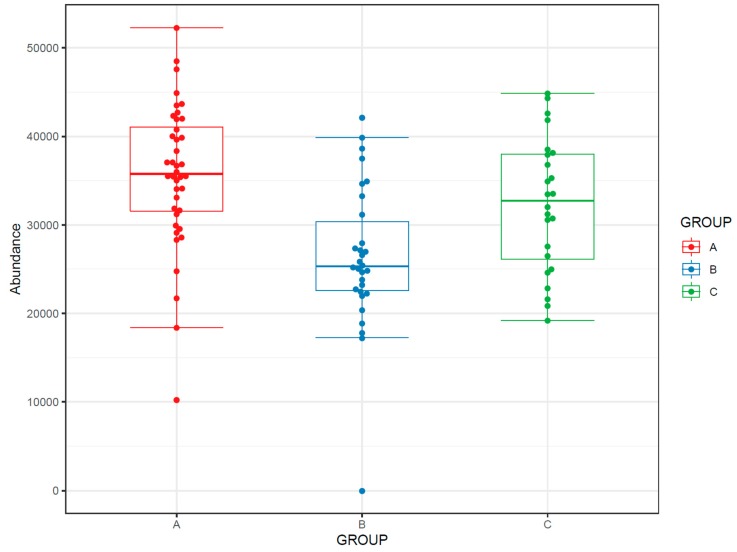
Beeswarm boxplots for a molecular feature subdivided into study group.

**Figure 15 metabolites-10-00135-f015:**
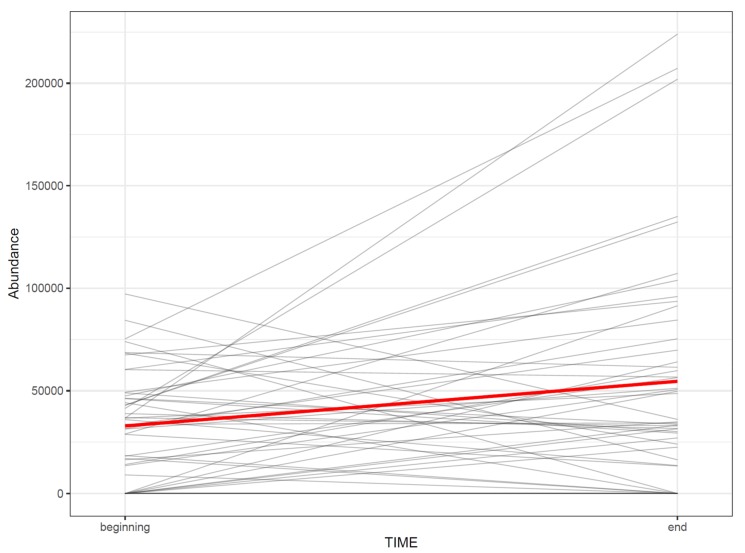
The change in the abundance of a molecular feature as a function of time in each subject. The thick red line represents the sample mean.

**Figure 16 metabolites-10-00135-f016:**
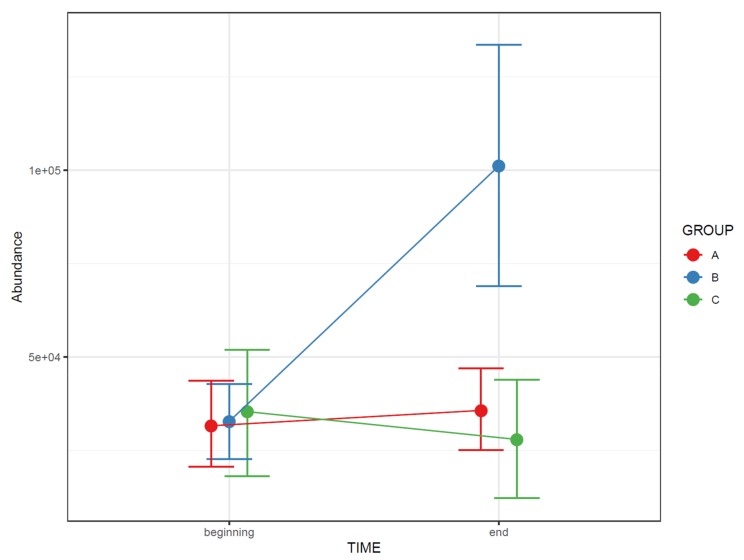
The change in the abundance of a molecular feature as a function of time in each study group. The whiskers depict 95% confidence intervals.

**Figure 17 metabolites-10-00135-f017:**
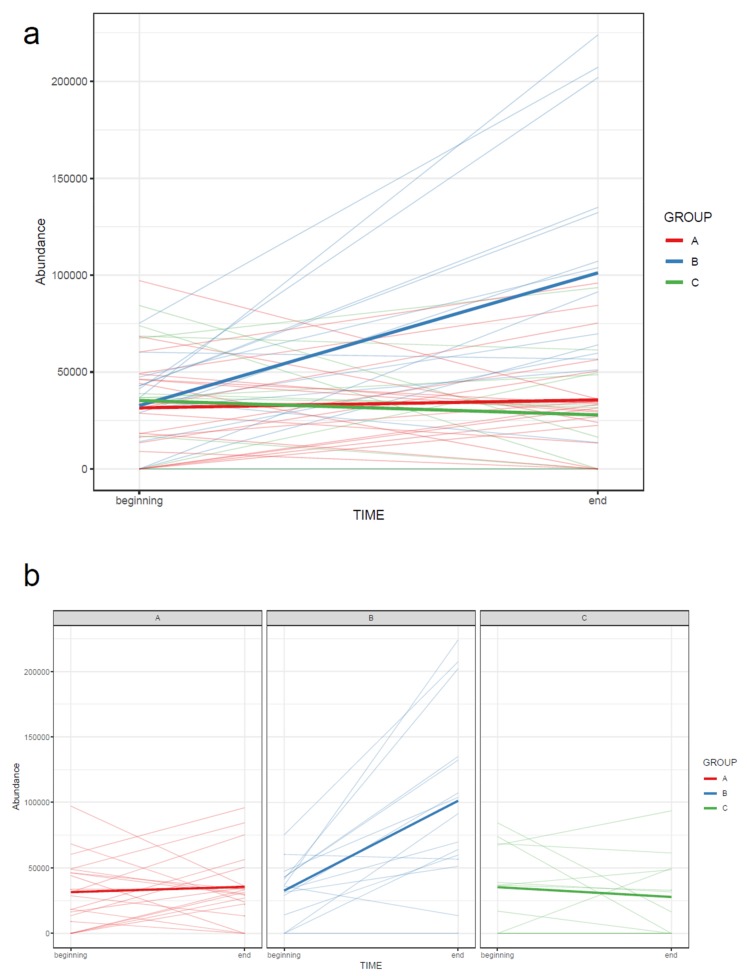
The change in abundance of a molecular feature between two time points in each subject, colored by group (**a**). Data with time series from multiple groups is easier to read when divided to small multiples (**b**). The bold lines represent group means. Note that the bold mean lines do not necessarily reflect an overall trend present in each subject.

**Figure 18 metabolites-10-00135-f018:**
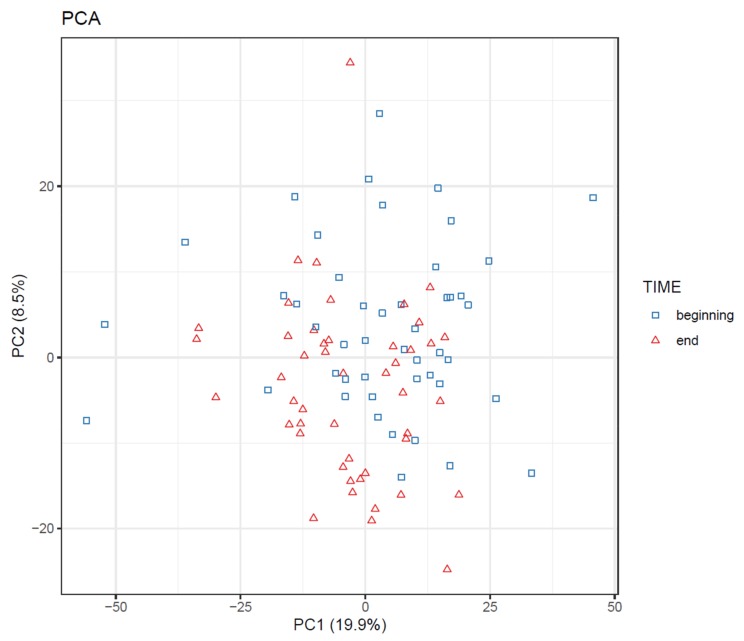
Principal component analysis (PCA) plot of samples from an intervention study, before and after the intervention. The time points are somewhat separated, but no clear clusters or outliers are visible.

**Figure 19 metabolites-10-00135-f019:**
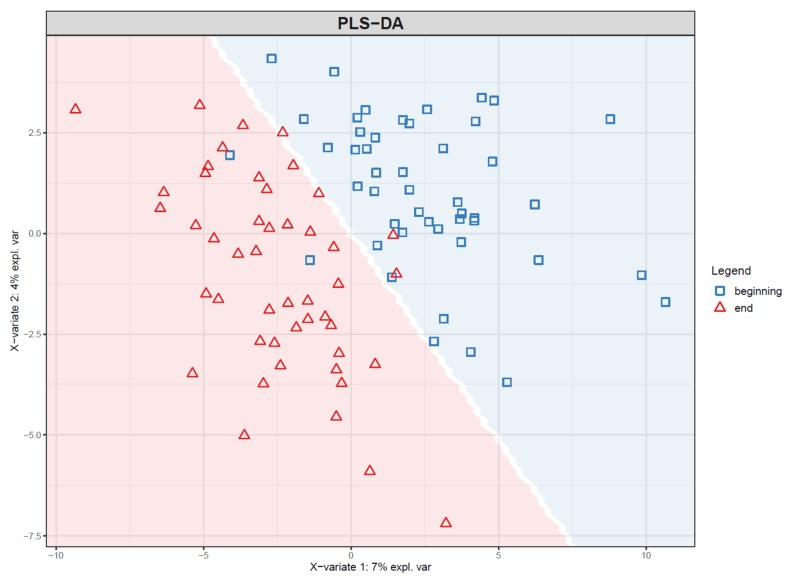
Score plot of the first two components of a partial least squares-discriminant analysis (PLS-DA) model trained to predict the time point of samples from an intervention study. The background color indicates the prediction of the model: samples in the blue area are classified to time point “beginning” and samples in the red area to time point “end”. Note that the time points are clearly more separated than in the corresponding principal component analysis (PCA) plot (Figure 18). This is to be expected, as PLS-DA finds components that specifically separate the two time points.

**Figure 20 metabolites-10-00135-f020:**
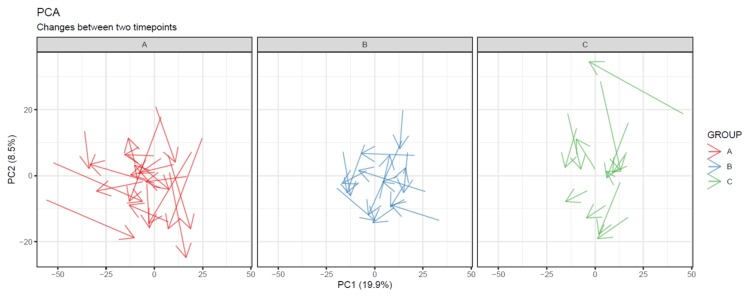
Changes in each subject between two time points visualized as arrows between points in a principal component analysis (PCA) plot. Samples in different groups are separated into subplots. While no group shows a systematic direction of change, we can observe that the subjects in group A show greater overall change that subjects in the other groups.

**Figure 21 metabolites-10-00135-f021:**
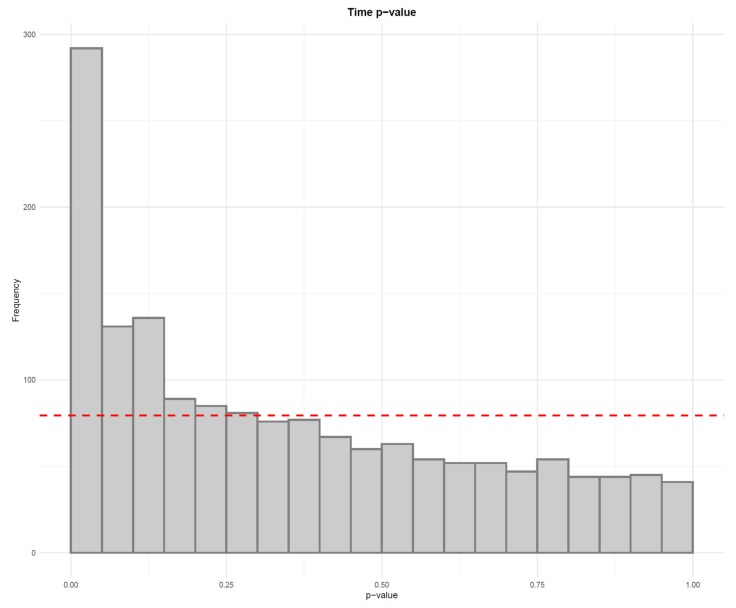
The distribution of *p*-values from linear models testing the difference in feature abundance between two time points. Since the distribution clearly deviates from the uniform distribution depicted by the red line, it can be argued that there is a true difference between the two time points.

**Figure 22 metabolites-10-00135-f022:**
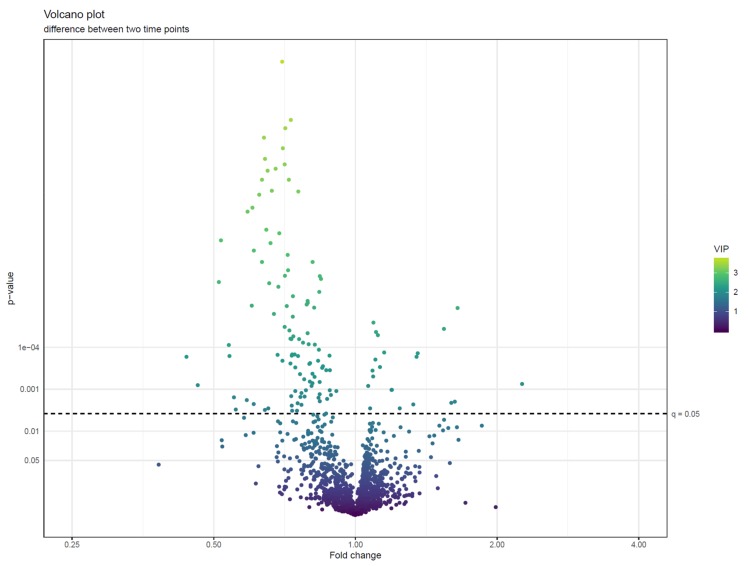
A volcano plot of *p*-values (negative log10 scale) from linear models testing the difference of feature abundances between two time points against fold changes between samples taken before and after a dietary intervention (log2 scale). The features are colored by variable importance in projection (VIP)-value from a partial least squares-discriminant analysis (PLS-DA) model trained to separate the two time points. We can observe that the features with the smallest *p*-values tend to have fold changes below 1, indicating that they are less abundant at the end of the intervention. Other metrics of effect size, like Cohen’s d values, can also be used in volcano plots.

**Figure 23 metabolites-10-00135-f023:**
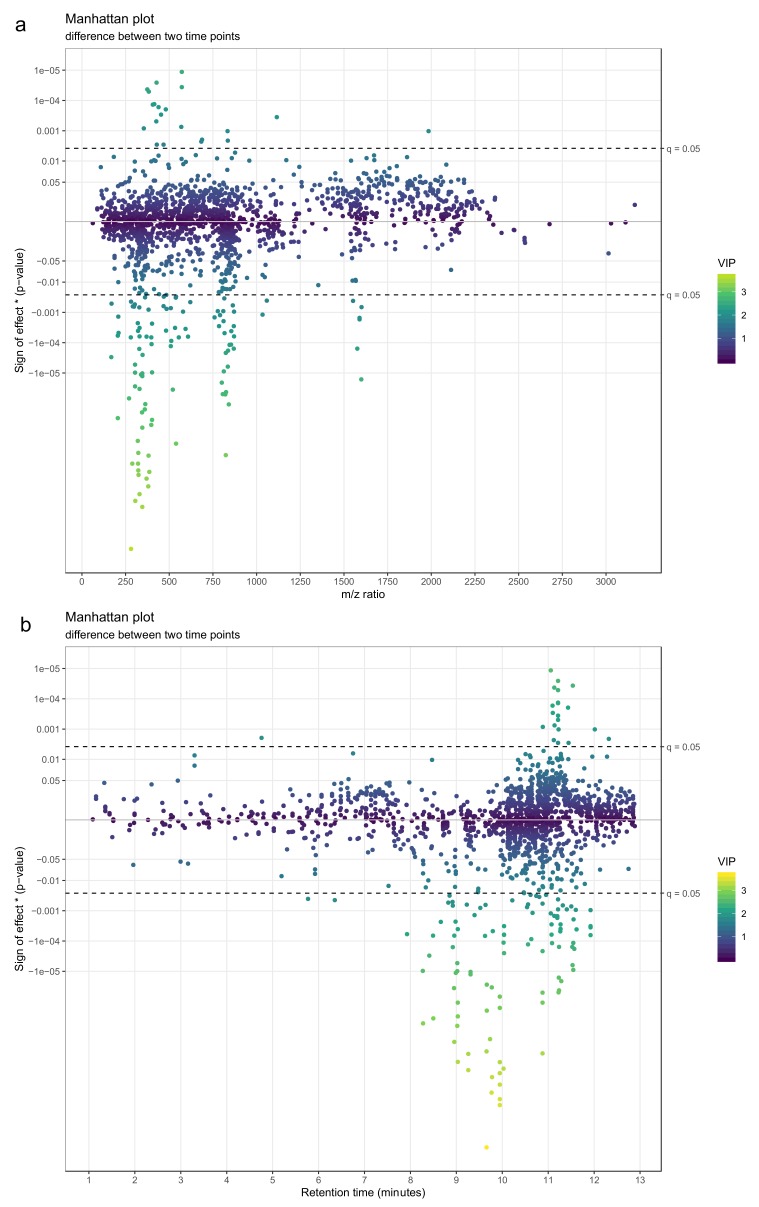
(**a**) A directed Manhattan plot of *p*-values from linear models testing the difference of feature abundances between two time points with mass-to-charge ratio of the features as x-axis. The points are colored by variable importance in projection (VIP)-value from a partial least squares-discriminant analysis (PLS-DA) model trained to separate the two time points. The most interesting groups of molecular features seem to have *m/z* ratios around 350 and around 800. Both groups are predominantly lower in the end of the intervention. (**b**) A similar directed Manhattan plot, only with retention time of the features as y-axis. The most interesting groups of molecular features seem to have retention times around 9–10 min and around 11 min. The first group is predominantly lower in the end of the intervention, while the features in the second group have mixed associations.

**Figure 24 metabolites-10-00135-f024:**
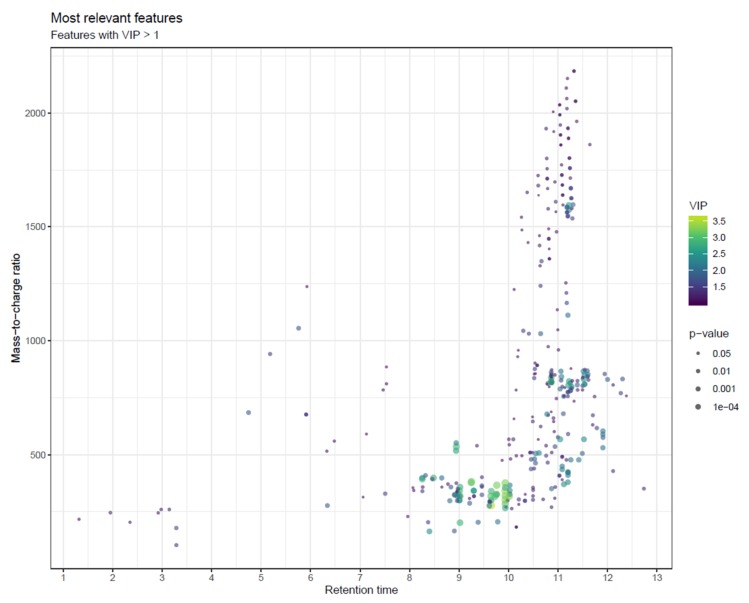
Scatter plot of molecular features in *m/z* vs retention time space, with the size of the points reflecting *p*-values from linear models testing the difference in feature abundances between two time points. The points are colored by variable importance in projection (VIP)-value from a partial least squares-discriminant analysis (PLS-DA) model trained to separate the two time points. To avoid too many overlapping points, only points with VIP value > 1 are drawn. We can observe that the most interesting group of features has retention times around 9–10 min and *m/z* ratios around 350.

**Figure 25 metabolites-10-00135-f025:**
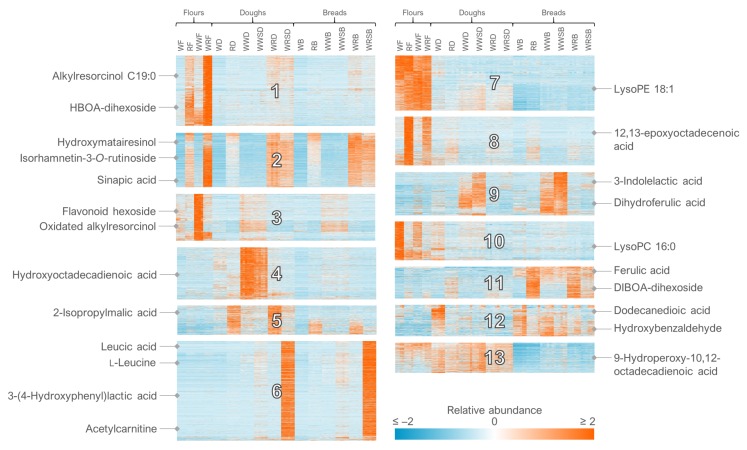
Heat map of all the 12,579 molecular features detected in reversed phase negative mode from cereal samples with some of the annotated compounds highlighted. *k*-Means clustering was applied to the dataset, dividing it into distinct clusters (*n* = 13) based on the relative abundance of the features across samples.

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
