# Peer review of "“Notame”: Workflow for Non-Targeted LC–MS Metabolic Profiling"

_metabolites, 2020, doi:10.3390/metabo10040135_

Round 1

Reviewer 1 Report

This is a nice protocol illustrating the untargeted LC-MS metabolomics workflow. In addition to the protocol, a tool for feature annotation is introduced. The paper is publishable but I have some concerns that should be addressed:

For the tool, I have the following comments:

Section 3.2.5. The authors claim this is a novel method for clustering. 'Clustering peaks' (pseudo-spectra extraction) by peak abundance or peak shape correlation is not a novel approach, but already reported in multiple papers (see Domingo-Almenara 2018, Anal. Chem. 90, 480-489). In fact, highly similar approaches based on graph/network methods were recently introduced (Senan et al 2019. 10.1093/bioinformatics/btz207, Kachman el al 2019, 10.1093/bioinformatics/btz798, or Kouril et al 2020, 10.1093/bioinformatics/btaa012). The authors should compare with one of these methods and describe how NoTaMe is different.

Also, lines 456-466 are nearly indecipherable. The description of the method is impossible to understand. The technical details can only be understood by a (bio)computational scientist and this seems to be a protocol for analytical chemists. Also, the relation between the computational and the analytical concepts that motivate this approach is not described (e.g., why a degree threshold?, what is the iteration to decompose subgraphs intended for?) It is impossible to understand what the tool is based on. Also, is it not defined if this is a chromatographic peak shape or feature abundance across samples correlation approach.

For the protocol, I have the following comments:

Is this HPLC or UHPLC? I assume that this is HPLC but the protocol does not specify or makes a distinction. It should clarify that if a UHPLC is used, an acquisition rate of 1.67scans/s is low. The scan rate should be at least 5 scans/s as recommended in the literature. 1.5 scans per second can barely profile a good peak shape for a 1-2 s peak in UHPLC, which then hampers the subsequent computational peak detection performance, and other analyses including peak-correlation analysis for annotation (e.g, CAMERA or even Notame).

Why the acquisition covers up to 1600 m/z? Is that necessary? It only adds more noise and unwanted features.

Line 283, 'missing values' is not defined, and there is no previous mention about it. The authors could add (see Section 3.2.4) after the "missing values" term is introduced.

3.2.3. Step 45. It's not mentioned from what samples are these features, I assume only QC samples are considered? Is this a method developed by the authors? If it is, where does it show to work better compared to existing approaches? Also, what is the rationale behind this approach? e.g., the authors say: "p-values should roughly follow the expected uniform distribution". Why?

Figure 5. Letter size is small and it is difficult to read.

Line 478 and section 3.3.1. it should read 'feature-wise (univariate)'

Author Response

Thank you for your comments and input, we have attached a word document addressing your points.

Kind Regards

M.Kokla

Reviewer 2 Report

The manuscript presents a protocol for the data acquisition and analysis in metabolomics studies. The steps described guide the user through a complete metabolomics analysis, comprising all the quality control and quality assurance necessary for unbiased, accurate and meaningful results. A number of software and R tools are used for the several stages of the analyses, and the notame package is also presented.

The authors aim to provide a protocol easy for other fellow scientist to adopt, and I think they have done a good job in putting together all the main quality controls and processing steps. There are some minor comments that I believe will improve clarity of the protocol and aid the future users of it.

General comments:

  1. the authors do not mention neither the inclusion of internal standards in the two sample preparation protocols, nor quantification of important metabolites in the data analysis. Considering the vast coverage of the metabolomics analysis here reported, I recommend the authors to include a paragraph regarding the best practice in compound quantification (and amend sample preparation accordingly).
  2. In several instances it is not clear to me how the described step was performed, as the procedure is described but the authors do not indicate any tool where it is implemented. It should be specified whether the steps are integrated in notame or were performed with other R packages or external software, and in all cases the specific function and parameters used should be described.This applies to:
    • drift correction, paragraph 3.2.2;
    • quality control, paragraph 3.2.3;
    • clustering molecular features originating from same metabolite, paragraph 3.2.5. Here an interesting approach for the grouping of features deriving from the same compound is described, and I think it is particularly important in this case giving additional information;
    • feature-wise analysis, paragraph 3.3.1.
  3. Given the variety of software and R packages used, I would like to see a schematic representation of the whole data processing pipeline indicating for each step the recommended tool(s), something along the lines of figure 2. Moreover, if in-house scripts were used, I would encourage the authors to share them in the supporting information. These two points are more a desire of the reviewer than a required correction for the manuscript, but I believe they will be incredibly beneficial for a reader trying to utilize the protocol.

Specific comments:

  1. LC-MS measurement, protocol point 28, Page 6 line 225: the authors recommend injecting quality control samples, described in the previous part as pools of all set of samples, every 12 real samples as well as at the beginning and end of the acquisition. How many QCs would they recommend injecting at the beginning of the acquisition for column conditioning, as well as at the end?
    Moreover, the only quality control samples mentioned are pools: could the authors explain why they decided not to include other kind of QCs, such as the widely used blanks?
  2. Peak picking and alignment, protocol point 30: the authors decided to use MS-DIAL for peak picking, but they should at least mention the existence of other equally valid and widely used software solutions for this step.
    Moreover, for some of the parameters used they provided a brief description and explanation on how to adjust them, while this is missing for some other (for instance, mass slice width of 0.1 Da). Please advise the user to always check their data and provide more information on how to adjust the parameters accordingly.
  3. Paragraph 3.2.4. Imputation, transformation, normalization and scaling: the authors mention that different techniques should be used for univariate and multivariate analyses and cite the paper of Di Guida and coworkers highlighting how the different data processing methods can affect the subsequent statistical analysis. However, only the steps to be followed prior to multivariate statistic are described. Do the authors follow the recommendation of Di Guida, i.e. “no missing value imputation, no scaling and no transformations are used prior to univariate statistical analysis”? If yes, please specify in text.
  4. Page 16, row 480: “Combine the features from the different analytical modes to a single data matrix”. I would like to see more information on how this step was performed, and on the problems associated with it. In particular, the author should mention the potential issues arising from both the different instrumental response in the two ionization modes, and metabolites detectable in both polarities, which will result in redundant features in the merged matrix.
  5. Page 29: Hierarchical clustering is performed with Multiple Experiment Viewer: could the author explain why they decided to use an external tool instead of R?
  6. Pag 31, line 796: “For both tools, HMDB or KEGG metabolite IDs are essential to avoid confusion from the multitude of nomenclature systems adopted different research groups” I agree with the authors on the use of HMDB and KEGG, but I would like them to explain better why they focus on these two databases and what they mean by the “multiple nomenclature system”.

Figures:

  1. Figure 13: please rearrange the labelling to improve readability, it is hard to understand what each label is referring to.
  2. Figure 14-17: remove labels above graph (for instance, “RP531_3663pos….” in figure 14).

Supporting information:

  1. The supporting information file has a paragraph describing study design on nutritional metabolomic analyses on humans, but I could not find any mention of this paragraph in the main manuscript.

Typos:

  1. Page 16 row 484 – 485: “Most typically parametric tests are used, but if the features do not satisfy the assumptions parametric tests, they may be replaced with non-parametric alternatives”. I think the authors meant something like “…but if the features do not satisfy the assumptions of parametric tests…”.
  2. Page 15, line 457: Pearson is strikethrough: was it meant to be deleted?
  3. Page 26, row 672: “Manhattan plots can be applied in metabolomics by using mass-to-charge ratio or retention time on the x-axis, the Manhattan plots can be used”. The repetition should be removed
  4. Page 31, line 793: “aetiology [1], [3](refs)”. Reference formatting should be corrected.

Author Response

(The authors gave the same response as above.)

Reviewer 3 Report

In this manuscript, Marietta and the coauthors proposed a detailed metabolomics workflow based on established metabolomics protocols and statistics. The protocol is well-written and comprehensive. it covers almost all the aspects involved in metabolomics, which will benefit researchers who want to use metabolomics in their biological studies. I have a few minor comments as follows.

  1. Step 6 of the protocol 1 (line 156 - 157), why QC samples only takes 10 ul plasma/serum but the analytical sample takes 100 ul. Is that a typo?
  2. In figure 8, please change x1, x2 to PC1, PC2. Please also comment on the difference between PCA and t-SNE, their advantage/disadvantage and which one should be chosen for better visualization.
  3. In Figure 11 a and b as well as in other figures, the figure legends are too small. Suggest to make them bigger and visible.
  4. Please comment on how to choose an appropriate correlation algorithm, Pearson vs. Spearman correlation.
  5. This protocol covers a lots of data processing and visualization through R programming. Please provide the R code and user protocol for running these analyses. Most biological researchers have limited knowledge of coding. So providing detailed instruction on how to use these R packages will be highly appreciated. Also please provide a small scale demo raw data so that users can test it.

Author Response

(The authors gave the same response as above.)

Round 2

Reviewer 1 Report

The authors have addressed all my concerns.